# Impact of Two Whole-Body Vibration Exercise Protocols on Body Composition of Patients with Metabolic Syndrome: A Randomized Controlled Trial

**DOI:** 10.3390/ijerph20010436

**Published:** 2022-12-27

**Authors:** Aline Reis-Silva, Ana C. Coelho-Oliveira, Elzi Martins-Anjos, Márcia Cristina Moura-Fernandes, Alessandra Mulder, Vinicius Layter Xavier, Vanessa A. Mendonça, Ana C. R. Lacerda, Laisa Liane Paineiras-Domingos, Redha Taiar, Alessandro Sartorio, Mario Bernardo-Filho, Danúbia C. Sá-Caputo

**Affiliations:** 1Medicina Laboratorial e Tecnologia Forense, Programa de Pós-Graduação Profissional em Saúde, Universidade do Estado do Rio de Janeiro, Rio de Janeiro 20550-013, RJ, Brazil; 2Laboratório de Vibrações Mecânicas e Práticas Integrativas, Instituto de Biologia Roberto Alcântara Gomes and Policlínica Universitária Piquet Carneiro, Universidade do Estado do Rio de Janeiro, Rio de Janeiro 20551-030, RJ, Brazil; 3Programa de Pós-Graduação em Fisiopatologia Clínica e Experimental, Faculdade de Ciências Médicas, Universidade do Estado do Rio de Janeiro, Rio de Janeiro 20550-170, RJ, Brazil; 4Programa de Pós-Graduação em Ciências Médicas, Faculdade de Ciências Médicas, Universidade do Estado do Rio de Janeiro, Rio de Janeiro 20550-170, RJ, Brazil; 5Departamento de Nutrição Aplicada, Instituto de Nutrição, Universidade do Estado do Rio de Janeiro, Rio de Janeiro 20550-170, RJ, Brazil; 6Departamento de Estatística, Instituto de Matemática e Estatística, Universidade do Estado do Rio de Janeiro, Rio de Janeiro 20550-000, RJ, Brazil; 7Programa de Pós-Graduação em Reabilitação e Desempenho Funcional, Universidade Federal dos Vales do Jequitinhonha e Mucuri, Diamantina 39100-000, MG, Brazil; 8Departamento de Fisioterapia, Instituto Multidisciplinar de Reabilitação e Saúde, Universidade Federal da Bahia, Salvador 40210-905, BA, Brazil; 9MATériaux et Ingénierie Mécanique (MATIM), Université de Reims, F-51100 Reims, France; 10Istituto Auxologico Italiano, IRCCS, Experimental Laboratory for Auxo-Endocrinological Research, 20145 Milan, Italy

**Keywords:** mechanical vibration, metabolic syndrome, body composition, exercise, adiposity

## Abstract

(1) Background: This study investigated the effects of two 6-week whole-body vibration exercise (WBVE) protocols on body composition in patients with metabolic syndrome (MSy). Thirty-three patients were allocated to either the Fixed Frequency WBVE Group (FFG-WBVE) or the Variable Frequency WBVE Group (VFG-WBVE). (2) Methods: A side-alternating vibration platform was used and the patients remained in the semi-squat position on this platform. In the FFG-WBVE (*n* = 12; median age = 50.50 years) and (body mass index BMI = 31.95 kg/m^2^), patients were exposed to 10 s of mechanical vibration at a fixed frequency of 5 Hz, followed by 50 s without vibration. In the VFG-WBVE (*n* = 10; median age = 57.50 years) and (BMI = 32.50 kg/m^2^), the patients performed 60 s of mechanical vibration at different frequencies from 5 to 16 Hz). Body composition evaluated through (bioelectrical impedance analysis and anthropometric measurements) were was evaluated before and after the all the interventions in each group. (3) Results: The VFG-WBVE decreased waist circumference, *p* = 0.01 and segmental fat mass [left arm, *p* = 0.01; right arm, *p* = 0.02 and trunk, *p* = 0.03]. Bone content increased, *p* = 0.01. No significant changes were observed in the FFG- WBVE. (4) Conclusions: In conclusion, this study demonstrated that 6 weeks of WBVE with a protocol with variable frequency can positively modify the body composition of MSY patients. These findings might contribute to improvements in the metabolic health of these patients.

## 1. Introduction

Metabolic syndrome (MSy) is characterized according to the International Diabetes Federation (IDF) by the presence of at least 3 factors, among: (i) elevated waist circumference (following ethnicity-specific values), (ii) elevated triglycerides or specific treatment for this lipid abnormality) (iii) reduced HDL cholesterol or specific treatment for this lipid abnormality, (iv) elevated blood pressure or treatment of previously diagnosed hypertension, or (v) elevated fasting glycemia or previously diagnosed type 2 diabetes mellitus [1].

MSy is related to unhealthy lifestyles such as smoking, increased consumption of a hypercaloric diet, and decreased physical activity, as well as being correlated with risk factors for the development of chronic non-communicable diseases such as, arterial hypertension, chronic kidney disease, cardiovascular disease, various types of cancer, and non-alcoholic steatosis [2]. A review conducted by Rochlani et al. reported that visceral adiposity appears to be the starting point for most of the mechanisms involved in MSy and is strongly associated with cardiometabolic complications [3].

Adipose tissue (AT) is distributed throughout the body in two main locations (subcutaneous and visceral fat) [4]. Longo et al. suggested that there is likely a genetic component regulating the ability to accumulate subcutaneous adipose tissue (SAT), if the ability to generate new adipocytes is impaired by genetic predisposition or stress (physiological and mental stress), the result is an expansion of the SAT with an increase in visceral adiposity and consequent increase in waist circumference (WC), which are both associated with increased morbidity and mortality [5,6].

Body mass index (BMI) has been considered the most used way to estimate body fat because it represents a simple parameter used to characterize overweight (BMI between 25 and 29 kg/m^2^) and obesity (BMI ≥ 30 kg/m^2^) according to the World Health Organization [7], However, although BMI is correlated with fat accumulation and metabolic health in large populations, it cannot be considered a sensitive index for differentiating lean mass from AT and it does not allow to determine the body fat distribution [8].

The distribution of AT, as well as lean mass, total water body, and bone mineral content (BMC), can be assessed using techniques that analyze body composition [9]. The gold standard for monitoring body composition is computed tomography (CT), nuclear magnetic resonance (NMR), and dual-energy X-ray absorptiometry (DEXA) [10]. However, studies suggest that the use of ionizing radiation (CT, DEXA), device dimensions, and high cost (NMR) reduce the availability of these techniques for research. In this context, a viable solution would be the analysis of body composition through the bioelectrical impedance analysis (BIA), as it is a cheaper technique, easy to apply, and does not use ionizing radiation [11]. Studies performed with BIA have shown a good correlation with other techniques for assessing body composition [12,13].

Although the benefits of regular physical exercise are well established, such as to reduce body mass and blood pressure, to raise high-density lipoprotein cholesterol, lower triglycerides, to improve insulin resistance and to delay the decline in physical fitness and body composition [14]. Studies describe the MSy patients have difficulty in adhering to a conventional physical exercise program due to increased body mass, musculoskeletal limitations, reduced physical fitness associated with lack of time, and lack of motivation [15,16].

To overcome these limits whole-body vibration exercise (WBVE), that is a systemic vibration therapy, may be, an useful alternative for the management of the patients, as they are easily accessible and have a low perception of effort [17,18]. Considering the MSy patients, Carvalho-Lima et al. demonstrated improvement in the quality of life; Sá-Caputo et al. reported an increase of the flexibility and decrease pain levels and Paineiras-Domingos et al. described positive responses in functional capacity [19,20,21].

In WBVE, the individual is exposed to mechanical vibrations produced by the vibrating platform (VP) [22]. A WBVE protocol should consider the biomechanical parameters (frequency, peak-to-peak displacement-PPD, and acceleration), positioning of the individual (static or dynamic postures), work time, rest time, number of sets, and number of sessions, according to the objectives specific to be achieved and the clinical condition of the individual [23,24].

During WBVE, mechanical vibration stimulates muscle spindles that respond with tonic vibration reflexes, thus increasing muscle activity [25]. In the literature, studies have identified a significant improvement in body composition in individuals submitted to WBVE. Pérez-Gómez et al. demonstrated improvements in body fat mass following a WBVE protocol in postmenopausal women [26]. Wei Deng also examined the effects of a WBVE training regimen on obese college students and observed a reduction in body fat percentage [27].

However, it is not well known yet what would be the best WBVE protocol to promote benefits related to body composition in MSy patients. The hypothesis is that WBVE may promote improvements in body composition parameters in MSy patients.

The aim of the current is to evaluate effects of two WBVE protocols: (i) fixed frequency (5 Hz) and (ii) variable frequency (5 to 16 Hz) on the body composition of MSy patients.

## 2. Materials and Methods

### 2.1. Study and Patients

Randomized, longitudinal and blinded analysis. Thirty-three patients with MSy diagnosis according to the IDF criteria were allocated into two WBVE groups: (i) Fixed Frequency Group WBVE(FFG-WBVE) or (ii) Variable Frequency Group WBVE (VFG-WBVE). The final sample consisted of 22 patients (16 women and 6 men), with a mean age of 56.36 years, and with BMI above 29 kg/m^2^. Patients were recruited in the Departamento de Cardiologia e Clínica Médica of the Hospital Universitário Pedro Ernesto, Universidade do Estado do Rio de Janeiro (HUPE/UERJ) and all the procedures were performed from October 2017 up to January 2020. All participants were informed about the study and signed the informed consent form. The WBVE interventions were performed in the Laboratório de Vibrações Mecânicas e Práticas Integrativas -LAVIMPI, Policlínica Universitária Piquet Carneiro, UERJ.

The study was approved by the Ethics Committee of the HUPE/UERJ (CAAE 54981315.6.0000.5259).

The same supervisor made the assessments of the patients, generated the random allocation and assigned the interventions. However, the data analysis was blinded.

The patients in the first evaluation signed the informed consent form, and various personal information were collected (age, gender, height, body mass, body mass index-BMI). Moreover, general questions about the physical activity, diabetes, hypertension, smoking, alcohol consumption, were answered, if yes or no.

### 2.2. Inclusion and Exclusion Criteria

Patients over 18 years of age, of both sexes and with diagnosis of MSy were included.

Exclusion criteria were hypertensive individuals with very high blood pressure levels with BP ≥ 180 × 110 mmHg; acute cardiovascular diseases in the last 6 months; patients who were under medications affecting the body mass and/or body composition; cardiac pacemaker or any neurological or muscle-skeletal diseases making impossible the individual to maintain stand position on the VP, and not acceptance of the informed consent.

The individuals included in the research were instructed no change the lifestyle (foods habits and physical activity) during the WBVE interventions.

### 2.3. Sample Size

The sample size was firstly calculated Miot considering the comparison of two groups to an infinite population [28]. The waist-to-hip ratio (WHR) was the quantitative variable used because is a strong risk factor for metabolic disorders, and it was considered a standard deviation of 0.06 and mean of 0.87 [29,30]. Thus, a sample size of eight patients in each group was determined. Following the same parameters (WHR), standard deviation and mean, the sample size was also calculated with the G-Power^®^ software (Franz Faul, Universitat Kiel, Germany) [30,31]. A power value of 80% and a value α of 0.05 for the WHR variable was considered, and the calculation identified the need of 18 patients, nine in each group.

### 2.4. Data Collection and Allocation

To minimize the risk of bias, opaque, sealed envelopes with two cards (one written FFG-WBVE and the other VFG-WBVE) were sequentially opened for randomization and allocation of individuals included in the respective groups. Patients were informed that two types of WBVE protocols would be tested.

### 2.5. Anthropometry

Physical examination included the determination of height, weight, WC, neck circumference (NC) and waist-hip ratio (WHR) by the same trained operators, according to the anthropometric standardization reference manual [32].

Standing height was determined by a stadiometer and weight was measured to the nearest 0.1 kg by using an electronic scale. WC was determined in standing position midway between the lowest rib and the top of the iliac crest after gentle expiration, with a non-elastic flexible tape measure B.

Ethnicity-specific WC cutoffs (men:90 cm, women:80 cm) were considered [33].

NC was measured with participants with the head erect and eyes facing forward, at the upper border of the laryngeal prominence [34]. The mean NC in individuals with normal body mass is 36.13 ± 2.14 cm in men and 31.59 ± 1.18 cm in women and the cutoff values obtained for NC to diagnose MSy is 37. 9 cm in men (sensitivity 0.875, specificity 0.640) and 34.0 cm in women (sensitivity 0.535, specificity 0.710) [35].

For WHR (WHR = WC (cm)/HC (cm)), hip circumference was measured at the largest point of the hip. WHR > 1 in men and > 0.8 in women are commonly used to diagnose increased accumulation of abdominal adipose tissue [36].

Body composition analysis (using BIA) was performed before the first session and after the last session of the WBVE interventions (see below).

### 2.6. Bioelectrical Impedance Analysis

Body composition was analyzed using a Tetrapolar Electrical Bioimpedance Analyzer (In Body 370, BIOSPACE, Republic of Korea) according to the manufacturer’s instructions. This analyzer processes fifteen impedance measurements using three different frequencies (5, 50, 500 kHz) on each of five body segments (right arm, left arm, trunk, right leg, left leg), and fifteen reactance measurements using tetrapolar 8—tactile point. The device provides separate body mass readings for different body segments, and uses an algorithm that incorporates age, height, and impedance to estimate total, and regional body fat mass, total and regional lean mass, total body water, skeletal muscle mass and. bone mineral content [37].

The measurement was performed in the morning between 7 and 10 am in a room with controlled temperature of 21 to 23 °C and the patients were instructed: (I) fasting for at least four hours; (II) not drinking coffee or alcohol in the last 48 h; (III) not performing intense physical exercise in the last 24 h; (IV) urinating at least 30 min before the measurement; (V) wear a bathing suit or light clothing; (VI) do not use any metal accessories; (VII) barefoot; (VIII) remain for at least 8–10 min in the position required by the device to minimize bias due to acute changes in body fluid distribution [38,39].

### 2.7. WBVE Interventions

Side-alternating VP (Novaplate, Fitness Evolution ^®^, São Paulo, Brazil) was used in both WBVE groups. The WBVE protocol took place for 6 weeks, twice a week (12 sessions) [40,41]. The patient was positioned barefoot, in a semi-squat position with 130° knee flexion controlled by a goniometer (Trident, Brazil), trunk erect, hands lightly resting on the side bar of the platform, relaxed shoulders and head in a neutral position. The WBVE was performed statically or dynamically (in alternating sessions). During the dynamic squat exercise, patients were asked to perform the squat slowly (2 s for eccentric and concentric phases).

Figure 1 Individual positioned and performing WBVE on a side alternating vibrating platform.

The peack to peack displacement (PPD) used in the protocols, in both groups, was indicated on the base of the VP (Figure 2), to each PPD, the distance in centimeters (cm) between the apices of the tibial malleolus was established: at 2.5 mm of PPD, the distance was of 3 cm; for 5.0 mm of PPD, the distance was 21 cm and for 7.5 mm of PPD, the distance was 39 cm.

In the (i) variable frequency group (VFG-WBVE, *n* = 16) the individual performed 1 min of WBVE (semi squat position) and 1 min of rest in 2.5-, 5.0- and 7.5-mm PPD. This sequence was performed 3 times from the 1st to the 4th session (18 min of total time); 4 times from the 5th to the 8th session (24 min of total time) and 5 times from the 9 to the 12th session (30 min of total time). The frequency increased from 5 to 16 Hz, with an increase of 1 Hz at each session.

In the (ii) fixed frequency group (FFG-WBVE, *n* = 17), the individual performed 1 min of WBVE (10 s with vibration and 50 s without vibration- semi squat position) and 1 min of rest) in 2.5, 5.0 and 7.5 mm PPD. This sequence was performed 3 times from the 1st to the 4th session (18 min of total time); 4 times from the 5th to the 8th session (24 min of total time) and 5 times from the 9 to the 12th session (30 min of total time). The frequency was 5 Hz in all sessions.

General and informal questions about undesirable effects, considering headache, nausea, dizziness, were asked during all stages of both interventions.

### 2.8. Statistical Analysis

Statistical analyzes were performed with the R program [42] and with the R packages: exactRankTests [43] and tableone [44]. Statistical tests were applied to compare the characteristics of participants of the two WBVE groups. For qualitative variables, expressed in terms of absolute values and percentage, Fisher’s exact test was applied. For quantitative variables, expressed in terms of median, first and third quartiles, the paired Wilcoxon signed-rank test was applied. Results were considered statistically significant if the *p*-value ≤ 0.05.

Efficacy was analyzed applying the intention-to-treat principle (ITT). Missing values were adjusted according to the mean value obtained, taking into account all variables and the total number of patients included in the sample [45].

## 3. Results

The Consolidated Standards of Reporting Trials flow diagram describing the steps of the enrollment of the participants is shown in Figure 3. Thirty-eight patients were recruited, five were excluded and thirty-three were randomized and allocated to the (*n* = 17 FFG-WBVE) and (*n*= 15VFG-WBVE) groups. A drop out in FFG-WBVE occurred due to health problems (*n* = 2) and personal reasons (*n* = 3) and in the group VFG-WBVE due to a loss of interest (*n* = 2), health problem (*n* = 1), work time mismatch (*n* = 3). Thus, twenty-two patients completed the protocol: (*n*= 12 FFG-WBVE) and (*n* = 10 VFG-WBVE).

The Table 1 shows the baseline anthropometric characteristics, the prevalence of concomitant diseases, and lifestyles of the two groups, respectively (FFG-WBVE vs. VFG-WBVE). No significant differences were found between the two groups for all the parameters.

The Table 2 shows the effects of the WBVE intervention on the body composition of patients in the two groups (FFG-WBVE vs. VFG-WBVE) before and after the interventions. A significant decrease in WC (*p* = 0.01) and an increase in bone content (*p* = 0.01) were found only in individuals in the VFG-WBVE group. Comparisons between the interventions (FFG-WBVE vs. VFG-WBVE) did not show significant changes. Table 2 also shows segmental fat values in various body segments before and after the two interventions. Although no significant changes were observed with FFG-WBVE, a significant decrease in segmental fat in the left arm (*p* = 0.01), right arm (*p* = 0.02), and trunk (*p* = 0.03) was found in the VFG-WBVE group. However, cross-group analyzes found no significant effects of WBVE. Finally, Table 2 demonstrates the values of segmental lean mass in various segments of the body before and after the interventions. No significant changes in segmental lean mass (kg) were detected in both groups and between groups for all parameters considered.

No undesirable effects, such as headache, nausea, dizziness, were not reported by the patients during all stages of both interventions.

## 4. Discussion

As hypothesized, the WBVE improved the body composition of MSy individuals. However, this finding was only observed in the VFG-WBVE, indicating the need of specific adjustment of biomechanical parameters. The results obtained in this work may favor the understanding of WBVE effects t and contribute to clinical practice.

A significant decrease in WC and an increase in BMC were found after 6 weeks of VFG-WBVE. In addition, a significant decrease in segmental fat mass (right, left arms and trunk) was found in MSy subjects of the VFG-WBVE. Considering the segmental lean mass in various body segments before and after the interventions, no significant changes were found in any of the groups (VFG-WBVE or FFG-WBVE). Therefore, the current study suggests that WBVE may have beneficial effects to modify some parameters of body composition in participants with MSy, and these findings are in agreement with Paley et al., reporting that regular and consistent physical exercise practices was able to reduce abdominal obesity and contribute to favorable changes in body composition [14].

In the current study, undesirable effects (headache, nausea, dizziness) related to WBVE interventions were not reported by the patients and, following this consideration, this type of exercise could be considered safe, viable and may be a good training option for people with difficulty in performing conventional exercises, as MSy patients [46,47].

According to Alberti et al., WC is a simple anthropometric measurement index associated with visceral adipose tissue and has been suggested as a predictor for MSy [1]. In VFG-WBVE, the WC decreased significantly (Table 2). Although there are some differences among the various protocols published, the finding of the current study is in agreement with (i) Song et al., who reported that the WBVE protocol (8 weeks, 22 Hz frequency, and 2 mm PPD) significantly decreased WC in obese postmenopausal women, (ii) Sañudo et al. also reported a significant decrease in WC with WBVE of 12 weeks, (frequency of 12 to 16 Hz and PPD of 4 mm) with dynamic and static exercises, in patients with type 2 diabetes mellitus [48,49]. In the present work, the decrease in WC observed with VFG-WBVE highlights the relevance of this type of exercise for the management of MSy.

Another finding of the current study was the reduction in the percentage of trunk segmental fat mass, which confirms the decrease in WC observed after VFG-WBVE and may suggest a reduction in visceral adipose tissue (Table 2). This result is in agreement with the results described by (i) Milanese et al., who reported significant improvement in trunk fat after 10 weeks of WBVE training, with frequency (from 40 to 60 Hz and PPD from 2.0 to 5.0 mm) in obese women, (ii) Severino et al., who found significant changes in fat percentage after 6 weeks of WBVE, three times a week (frequency from 25 to 40 Hz and 2 mm amplitude) in obese postmenopausal Hispanic women and (iii) Oh et al. demonstrated a decrease in visceral fat mass with 6 months of WBVE protocol (frequency from 30 to 50 Hz with high and low amplitude) in patients with non-alcoholic fatty liver disease [50,51,52]. In contrast, Klarner et al., comparing 2 protocols with fixed-frequency WBVE for 24 months, in two types of platforms (vertical, 35 Hz, 1.7 mm; and alternating platform 12.5 Hz, 12 mm), found no effects in body fat mass in postmenopausal women [53]. The differences among these studies can be attributed to the type of protocol used. In fact, Milanese et al., Severino et al., Oh et al. used an increasing frequency (Hz), while Klarner et al. [54] performed a fixed frequency (Hz) [50,51,52,53].

The As far as AT is concerned, the current study also found a significant decrease in segmental fat mass in the right and left arms in the VFG-WBVE group (Table 2). Although some studies have shown a significant decrease in total body fat mass percentage using WBVE (Oh et al. [52]; Munñoz et al. [54]; Severino et al. [51]), few studies evaluated segmental fat. In agreement with our results, Agüero et al. found a decrease in upper limb segmental fat mass with a WBVE protocol of 20 weeks of squatting exercise on a vertical platform (frequency from 25 to 30 Hz and 2 mm of amplitude) in adolescents with Down syndrome [55].

Paley et al. reported that intra-abdominal fat is a long-term predictor of insulin resistance and a critical mechanism between MSy and cardiovascular disease, concluding that exercises capable of decreasing abdominal adiposity may play a relevant role in the management of metabolic diseases, including obesity and related disorders [14].

Another finding of the current work was representd by the increase in BMC in the VFG-WBVE (Table 2). According to our finding, Erceg et al. reported a significant increase in BMC with the WBVE protocol (3 times a week for 10 weeks, frequencies 30 to 40 Hz) in overweight Latino boys [56]. On the other hand, Gómez-Cabello found no effects on bone mineral density and BMC in non-institutionalized elderly after WBVE protocol, 3 times a week for 11 weeks, frequency of 40 Hz and amplitude of 2 mm [57]. Probably, the different responses may be related to the different protocols used and the populations studied, since Gómez-Cabello evaluated only elderly, while in the current syudy and in the Erceg et al. were evaluated younger populations [56,57].

Considering body mass, BMI, NC, WHR, total body water and muscle mass in (Table 2), no significant differences were found after WBVE interventions. In relation to muscle mass and according to our results, Rubio-Arias et al. also found no effects on muscle mass after 6 weeks of WBVE in 40 healthy young male adults, with frequency (from 30 to 45 Hz and amplitude from 2 mm to 4 mm) [58]. In contrast, Chang et al. found significant changes in the muscle mass index through a WBVE protocol, 3 times a week, performed for three months with a vibration frequency (12 Hz and 3 mm amplitude) in the elderly and Bogaerts et al. also demonstrated a significant improvement in muscle mass in the elderly performing WBVE for a maximum of 40 min for 12 months (frequency from 30 to 60 Hz) with static and dynamic exercises [59,60]. It may be relevant to point out that a longer WBVE protocol, 3 months (Chang et al.) and 12 months (Bogaerts et al.) could be more effective in promoting changes in muscle mass than a short protocol of period used in the study by (Rubio-Arias et al.) and in the current study with only 6 weeks [58,59,60]. Although it has been reported that TBW and extracellular body water is generally increased in obese women and in patients with sarcopenic obesity [61], the interventions with WBVE did not significantly change TBW. However, the effects on TBW during WBVE could probably by influenced by the daily fluid intake (not recorded) of the patients during the entire period of the study, thus preventing in the present study to draw definitive conclusion on this specific aspect [62].

Regarding the segmental lean mass in the different segments (right arm, left arm, trunk, right leg and left leg), no significant changes were detected (Table 2) in MSy patients using protocols with WBVE with fixed or variable frequencies. This finding is in agreement with (i) Marín-Cascales et al., who reported no significant changes in total lean mass after 24 weeks with 3 weekly sessions of WBVE (frequency from 35 to 40 Hz and 4mm amplitude) in postmenopausal women [63]. (ii) Rubio-Arias et al. also did not report significant improvements in segmental lean mass of the trunk, right leg, and left leg after 6 weeks of WBVE (frequency30 to 45 Hz and 4 mm amplitude) in healthy individuals [64].

## 5. Strength

The strength of this investigation is that, to the best of our knowledge, no previous studies evaluated the effects of WBVE on body composition in patients with MSy. The WBVE-induced improvements in body composition of individuals with MSy may contribute to reduce cardiovascular risks and to improve the management of these individuals.

## 6. Conclusions

The current study demonstrated that 6-weeks of VFG-WBVE, performed actively, can positively modify body composition in individuals with MSy. The improvement in fat mass, on the left and right arms and trunk, isclinically noteworthy, since the reduction of this fat tissue can contribute to improvements in metabolic health and reduce the cardiovascular risk factor. However, further additional studies are requested to determine whether higher doses of vibration, WBVE combined with other exercises on the platform, or a longer duration of training (for example, 24 weeks) can promote best effects on the body composition of MSy patients.

## Figures and Tables

**Figure 1 ijerph-20-00436-f001:**
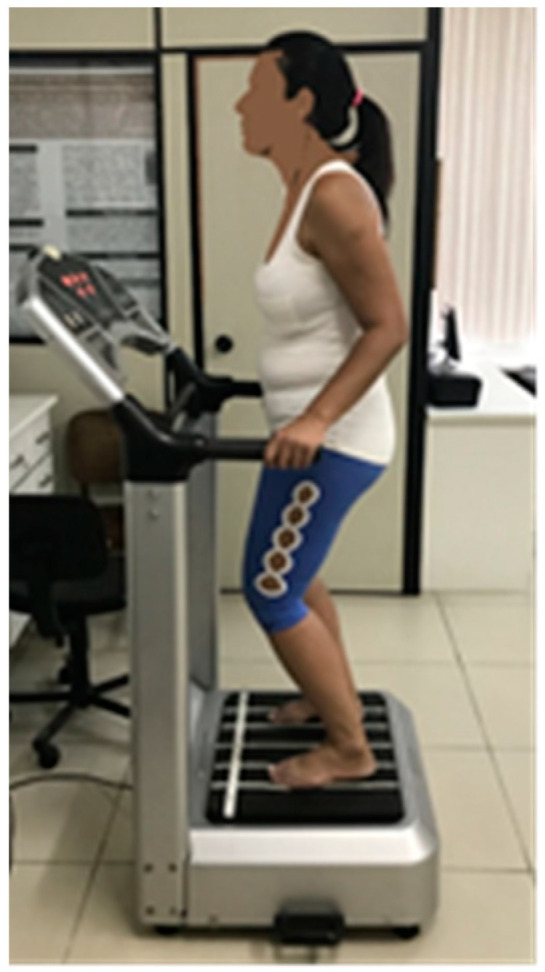
Patient positioned and performing WBVE on a side-alternating vibrating platform.

**Figure 2 ijerph-20-00436-f002:**
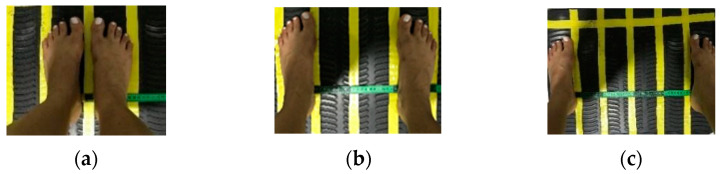
The position of peak-to-peak displacement (PPD) was used and the distance, in cm, between the feet at the base of VP was to (**a**) 2.5 mm of PPD, 3 cm between the feet; (**b**) 5.0 mm of PPD, 21 cm between the feet; and in (**c**) 7.5 mm of PPD, 39 cm between the feet.

**Figure 3 ijerph-20-00436-f003:**
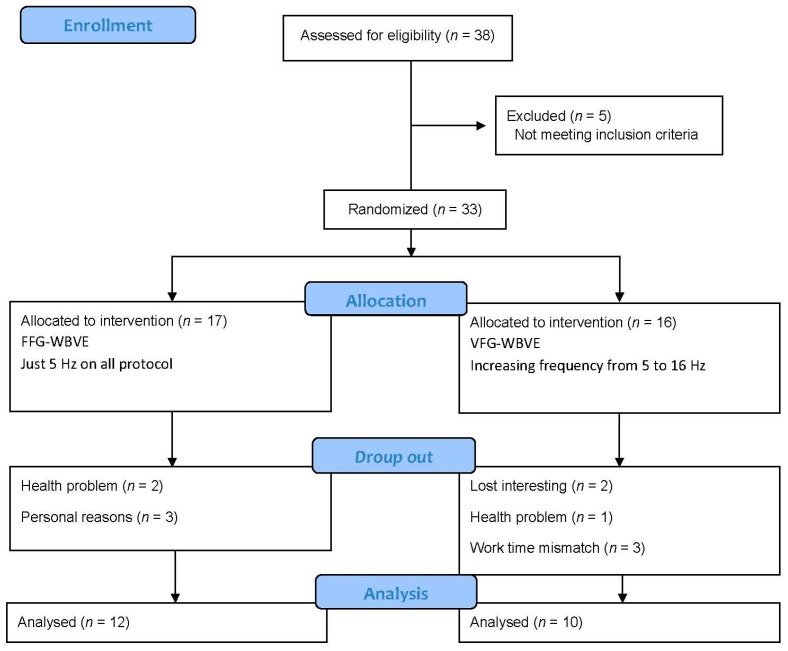
Flow of patients through the intervention.

**Table 1 ijerph-20-00436-t001:** Some anthropometrics characteristics and prevalence of concomitant diseases and lifestyle of groups (FFG-WBVE vs. VFG-WBVE).

Variables	FFG-WBVE	VFG-WBVE	
	Baseline	Baseline ^a^	*p*
Body mass (kg)	90.45 (77.10;101.00)	83.25 (70.70;87.65)	0.62
Height (m)	1.62 (1.56;1.69)	1.56 (1.55;1.61)	0.44
Age (years)	50.50 (46.75;62.25)	57.50 (49.75;67.00)	0.44
WHR	0.96 (0.93;1.02)	0.99 (0.98;1.01)	0.5
BMI (kg/m^2^)	31.95 (29.08;36.07)	32.40 (30.80;36.53)	0.81
WC (cm)	105.40 (102.01;113.85)	107.75 (105.25;110.70)	0.41
NC (cm)	37.50 (35.58;38.08)	38.18 (37.50;38.88)	0.27
Diabetes (%)	6(54.5)	4 (40.0)	0.67
Hypertension (%)	12 (100.0)	10 (100.0)	1.00
Alcohol consumption (%)	6 (54.4)	2 (20.0)	0.18
Physical exercises (%)	3 (27.3)	5 (50.0)	0.38
Smoking (%)	0 (0.0)	2 (20.0)	0.21

HR (waist to hip ratio); BMI (body mass index); WC (waist circumference); NC (neck circumference); FFG-WBVE (fixed frequency group-whole body vibration exercises); VFG-WBVE (variable frequency group-WBVE) vibration exercise); WBVE (whole- body vibration exercise). ^a^-Median (1st; 3rd quartiles) a-absolute frequency (percentage).

**Table 2 ijerph-20-00436-t002:** Body composition obtained with BIA of the two groups (FFG-WBVE vs. VFG-WBVE) before and after interventions.

	FFG-WBVE			VFG-WBVE		
Variables	Before ^a^	After ^a^	*p*	Before ^a^	After ^a^	*p*
Body mass (kg)	90.45 (77.10;101.10)	90.00 (78.17;101.10)	0.55	83.25 (77.70;87.65)	83.30 (77.78;85.95)	0.12
BMI (kg/m^2^)	31.95 (29.07;36.08)	32.60 (28.9;36.48)	0.71	32.40 (30.80;36.52)	33.15 (31.02;35.60)	0.28
NC (cm)	37.50 (35.15;38.15)	37.00 (33.50;38.75)	0.98	38.15 (37.50;38.88)	38.30 (36.80;39.00)	0.19
WC (cm)	105.4 (102.0;113.8)	104.75 (98.95;113.25)	0.3	107.8 (105.4;110.7)	105.5 (101.8;108.2)	0.01 *
WHR	0.96 (0.93;1.01)	0.96 (0.93;1.03)	0.58	0.99 (0.97;1.01)	1.00 (0.97;1.01)	0.8
Total body water (kg)	36.05 (32.95;40.65	36.15 (32.80;43.33)	0.86	31.95 (30.60;34.91)	32.40 (31.02;34.88)	0.77
Lean mass (kg)	46.45 (42.12;52.30)	46.55 (42.05;55.73)	0.98	41.05 (39.30;41.05)	41.55 (39.75;44.83)	0.78
Body fat mass (kg)	35.25 (29.38;43.20)	35.40 (29.18;42.10)	0.63	37.50 (34.20;41.12)	37.75 (33.08;40.95)	0.19
Bone content (kg)	2.74 (2.54;3.05)	2.64 (2.64;3.57)	0.69	2.52 (2.34;2.72)	2.62 (2.405;2.860)	0.01 *
Fat free mass (kg)	49.20 (44.85;55.33)	49.30 (44.67;58.95)	0.95	43.55 (41.77;47.02)	44.00 (42.27;47.50)	0.58
Skeletal muscle mass (kg)	27.50 (24.32;31.07)	27.50 (24.20;33.17)	0.94	23.80 (22.52;26.10)	24.05 (22.98;26.45)	0.53
**Segmental Fat Mass**						
Left arm (%)	48.95 (43.83;54.75)	48.15 (43.58;54.67)	0.41	52.60 (52.33;53.98)	51.80 (50.90;53;48)	0.01 *
Right arm (%)	47.90 (42.88;54.15)	48.90 (41.05;54.67)	0.66	52.30 (51.85;52.88)	51.60 (50.55;53.00)	0.02 *
Trunk (%)	40.55 (39.48;47.02)	41.70 (38.45;47.08)	0.45	45.70 (44.83;46.77)	44.65 (43.20;46.90)	0.03 *
Left leg (%)	40.00 (34.33;45.00)	40.55 (33.25;46.67)	0.62	43.10 (42.60;44.15)	41.75 (38.90;44.02)	0.08
Right leg (%)	39.55 (34.48;44.67)	39.30 (35.15;47.25)	1	43.40 (42.38;44.40)	42.10 (38.83;43.48)	0.08
**Segmental Lean Mass**						
Left arm (kg)	2.82 (2.35;3.12)	2.76 (2.36;2.76)	0.66	2.40 (2.21;2.78)	2.44 (2.27;2.88)	0.22
Right arm (kg)	2.79 (2.32;3.260)	2.76 (2.39;3.52)	0.45	2.37 (2.23;2.89)	2.42 (2.32;2.92)	0.57
Trunk (kg)	23.10 (20.93;25.70)	23.00 (21.20;26.82)	0.58	20.10 (23.50;19.70)	20.75 (20.12;23;55)	0.19
Left leg (kg)	7.54 (6.38;9.29)	7.47 (6.39;8.96)	0.63	6.59 (6.24;7.12)	6.43 (6.23;7.04)	1
Right leg (kg)	7.58 (6.84;9.31)	7.52 (6.80;9.08)	0.83	6.61 (6.25;7.05)	6.53 (6.22;6.96)	0.6

FFG-WBVE (fixed frequency group-WBVE); BIA (bioelectral impedance analysis); VFG-WBVE (variable frequency group-WBVE); WBVE (whole-body vibration exercise); BMI (body mass index); NC (neck circumference); WC (waist circumference); WHR (waist-to-hip ratio); ^a^-Median (1st; 3rd quartiles) * *p* ≤ 0.05.

## Data Availability

Not applicable.

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
