# Peer review of "Impact of Two Whole-Body Vibration Exercise Protocols on Body Composition of Patients with Metabolic Syndrome: A Randomized Controlled Trial"

_ijerph, 2022, doi:10.3390/ijerph20010436_

Round 1

Reviewer 1 Report

Due to the unclear description of research procedure and methods, some content need to be further revised. Please see below for more details.

1. The abstract

The age and main characteristics of subjects has not been described.

The procedure description of the intervention measures was unclear.

The results and conclusion were not clear. The result should include clear data and statistical value. And the conclusion should be the summarization of the main results.

2. Introduction

In the introduction part, the research progress and current situation on the influence of exercise intervention methods on diseases have not been fully elaborated

3. Method

The main characteristics of subjects has not been described.

Whether the level of physical activity is taken into account.

The description of sample size calculation is not detailed, and the calculated sample size result is inconsistent with the actual sample size

How to ensure the quality of research due to the high rate of lost visits

3 Discussion

The discussion section need more comparison and analysis with other similar research results.

Author Response

  • The abstract
  1. The age and main characteristics of subjects has not been described.

Thank you. The Information regarding the age and main characteristics of patients were inserted in the abstract.

  1. The procedural description of the intervention measures was not clear.

Thank you. Adjustments have been made to the text to clarify the intervention measures.

  1. The results and conclusion were not clear. The result must include clear data and statistical value. And the conclusion should be the summary of the main results.

Thank you. The results in the abstract were rewritten, including the statistical values and the conclusion summarized the main results.

  • Introduction

 1. In the introduction part, the research progress and current situation on the influence of exercise intervention methods on diseases have not been fully elaborated

Thank you. In the introduction, information was added regarding research progress in the WBVE and the influence of physical exercise on these patients.

Method

  1. The main characteristics of subjects has not been described.

Thank you. The main characteristics of the patients were added in the materials and methods section in the item “study and patients”.

 2. Whether the level of physical activity is taken into account.

Thank you. All patients included in the study answered general questions about lifestyle and reported about the physical exercise practice. The data are shown in table 1.

  1. The description of sample size calculation is not detailed, and the calculated sample size result is inconsistent with the actual sample size

Thank you. Additionally, the calculation of the sample was performed using the G-Power® software (Franz Faul, Universitat Kiel, Germany) and it was included in the study.

  1. How to ensure the quality of research due to the high rate of lost visits

Thank you. To ensure the quality of the research, the intention-to-treat principle was applied.

Discussion

  1. The discussion section need more comparison and analysis with other similar research results.

Thank you. Some sentences in the discussion section were rewritten to clarify the comparison of the current study with information of the cited publications.

Reviewer 2 Report

General: 

The authors investigated the impact of two whole body vibration exercise protocols on body composition on patients affected by Metabolic Syndrome. The results show that 6 weeks of treatment can positively modify body composition parameters and these findings could improve the metabolic health of the individuals.

Major comments:

Abstract:

The conclusion section should be fixed. You should merge the presence of point a) and b).

General comments: 

The style of the bibliography should be inserted according to the guidelines for authors.

Choose how to write “Figure” into the text. In fact, sometimes is written as “Figure” and sometimes as “Fig.” (line 236).

The discussion section should be more concise and include more reflections on the data obtained.

Conclusion: please add the limits and the advantages of passive and active physical exercise.

Line 367: management: what kind of management? Pharmaceutical, clinical…? Please define better this point.

Minor comments:

Introduction:

Ref 1: please check that this reference is the most recent of IDF

Line 56: please replace “serious disease” with “chronic non-communicable diseases”

Line 56: please delete “stroke”, “heart” and add “arterial hypertension, chronic kidney disease, cardiovascular disease” see this reference: 

Line 57: “A review conducted by Rochlani et al. ….”

Line 59: MSy “onset”

Line 67: please change in “to an increased mortality”

Line 69: “Although the body mass index… “should be wrapped

Line 71: “organization. Body fat percentage…”

Line 74: “population, but it is not very sensitive for measuring body fat distribution”

Line 80: “In this sense” replace with “in this context”

Line 81: replace with “bioelectrical impedance analysis (BIA)

Line 88: “In this context” should be replaced with “To overcome these limits”. Moreover, the sentence should not break the line.

Line 92: and through the whole manuscript, when the term is referred to the people studied the word “individual” should be changed with “patients”

Matherials and Methods:

Line 111: “Study and individuals” change in “study and patients”

Line 133: Please add “not acceptance of informed consent”

Line 136: exercise profile change in “physical activity”

WBVE Interventions:

Line 197: Please start the sentence with the capital letter. Change with: “Figure 1. The exercise”

Results:

Line 244: “Table 1 shows”

Author Response

Major comments:

Abstract:

The conclusion section should be fixed. You should merge the presence of point a) and b).

Thanks. The summary was improved and we tried to make the conclusion section more clear

 General comments: 

The style of the bibliography should be inserted according to the guidelines for authors.

Thank you.

We reviewed the bibliography and are available for further adjustments.

Choose how to write “Figure” into the text. In fact, sometimes is written as “Figure” and sometimes as “Fig.” (line 236).

Thank you.

Yes, we left the information homogeneous throughout the text.

The discussion section should be more concise and include more reflections on the data obtained.

Thank you. The discussion session was modified to be more concise and improve reflections on the data obtained.

Conclusion: please add the limits and the advantages of passive and active physical exercise.

Thank you. Our patients performed active exercise (squatting) in both groups (FFG-WBVE and VFG-WBVE).

Line 367: management: what kind of management? Pharmaceutical, clinical…? Please define better this point.

Thank you. Some terms have been replaced for a better understanding of the text.

Minor comments:

Introduction:

Ref 1: please check that this reference is the most recent of IDF

Line 56: please replace “serious disease” with “chronic non-communicable diseases”

Line 56: please delete “stroke”, “heart” and add “arterial hypertension, chronic kidney disease, cardiovascular disease” see this reference: 

Line 57: “A review conducted by Rochlani et al. ….”

Line 59: MSy “onset”

Line 67: please change in “to an increased mortality”

Line 69: “Although the body mass index… “should be wrapped

Line 71: “organization. Body fat percentage…”

Line 74: “population, but it is not very sensitive for measuring body fat distribution”

Line 80: “In this sense” replace with “in this context”

Line 81: replace with “bioelectrical impedance analysis (BIA)

Line 88: “In this context” should be replaced with “To overcome these limits”. Moreover, the sentence should not break the line.

Line 92: and through the whole manuscript, when the term is referred to the people studied the word “individual” should be changed with “patients”

Thank you. The reference to the International Diabetes Federation was updated in the text, and the other suggestions regarding terms and collocations were followed in the work.

Matherials and Methods:

Line 111: “Study and individuals” change in “study and patients”

Line 133: Please add “not acceptance of informed consent”

Line 136: exercise profile change in “physical activity”

Thank you. All suggestions were followed throughout the text. 

WBVE Interventions:

Line 197: Please start the sentence with the capital letter. Change with: “Figure 1. The exercise”

 Thank you. Correction performed.

Results:

Line 244: “Table 1 shows”

Thank you. Correction performed.

Reviewer 3 Report

The manuscript describes the effects of short-term whole-body vibration exercise on body composition and anthropometric characteristics. I find this very relevant due to the paucity of sound scientific data on this issue. Such an intervention could bring significant benefits to metabolic syndrome patients.

I would expect the authors to comment on the numerical increase in body mass index and body water, even though statistically non-signficant. Do they find this as relevant? 

Given that the differences for segmental fat mass are very small, I would question if the authors took into account the variability for measurements (especially those regarding the body composition) and if they assessed any internal (during the study) variation coefficient.

Author Response

The manuscript describes the effects of short-term whole-body vibration exercises on body composition and anthropometric characteristics. I find this very relevant due to the paucity of sound scientific data on this issue. Such an intervention could bring significant benefits to metabolic syndrome patients.

We agree and we thank you for these considerations.

  1. Would expect the authors to comment on the numerical increase in body mass index and body water, even though statistically non-significant. Do they find this as relevant?

We agree and add a paragraph in the discussion related to this numerical but non-significant increase.

  1. Given that the differences for segmental fat mass are very small, I would question if the authors took into account the variability for measurements (especially those regarding the body composition) and if they assessed any internal (during the study) variation coefficient.

Thank you. We agreed and the coefficient of internal variation related to the percentage of segmental fat (left arm right arm, trunk, left leg, and right leg), and the results suggest homogeneity of the data, as shown in the table below.

CV

FFG- WBVE

VFG- WBVE

Variables

Beforea

Aftera

Beforea

Aftera

Left arm (%)

 17.17

 18.32

 7.22

 7.05

Right arm (%)

 16.56

 19.72

 7.19

 7.51

Trunk (%)

 12.78

 15.86

 7.38

 8.07

Left leg (%)

 20.68

 20.78

 12.77

 12.73

Right leg (%)

 20.76

 21.80

 12.56

 12.64

Round 2

Reviewer 2 Report

Thank you for your corrections. Now your manuscript is ok.